# Context congruence: How associative learning modulates cultural evolution

**Monica Tamariz**[1,2], **Aliki Papa**[1,3]*, **Mioara Cristea**[1], **Nicola McGuigan**[4]

**1** Heriot-Watt University, Edinburgh, Scotland, United Kingdom, **2** Edinburgh Napier University, Edinburgh, Scotland, United Kingdom, **3** University of Bergen, Bergen, Norway, **4** University of the West of Scotland, Paisley, Scotland, United Kingdom

* Aliki.Papa@uib.no

**Data Availability Statement:** Data are available from the Zenodo repository: https://zenodo.org/record/6463741#.YmlUuVDMJS7.

## Abstract

The adoption of cultural variants by learners is affected by multiple factors including the prestige of the model and the value and frequency of different variants. However, little is known about what affects onward cultural transmission, or the choice of variants that models produce to pass on to new learners. This study investigated the effects on this choice of congruence between two contexts: the one in which variants are learned and the one in which they are later transmitted on. We hypothesized that when we are placed in a particular context, we will be more likely to produce (and therefore transmit) variants that we learned in that same (congruent) context. In particular, we tested the effect of a social contextual aspect–the relationship between model and learner. Our participants learned two methods to solve a puzzle, a variant from an "expert" (in an expert-to-novice context) and another one from a "peer" (in a peer-to-peer context). They were then asked to transmit one method onward, either to a "novice" (in a new expert-to-novice context) or to another "peer" (in a new peer-to-peer context). Participants were, overall, more likely to transmit the variant learned from an expert, evidencing an effect of by prestige bias. Crucially, in support of our hypothesis, they were also more likely to transmit the variant they had learned in the congruent context. Parameter estimation computer simulations of the experiment revealed that congruence bias was stronger than prestige bias.

## 1 Introduction

Human culture is the unique product of cumulatively adaptive evolution [1, 2], which has led to diversity and sophistication levels unparalleled in the animal kingdom [3]. The social transmission of cultural traits has been extensively studied (see reviews by [4–7], e.g., in the laboratory [8–11], with mathematical models [12–15], agent-based computer simulations [16, 17], and even in real-world paradigms [18, 19]. We now have a good understanding of the many biases that influence which of the cultural variants that learners observe will be adopted [20–23]. In contrast, the factors affecting which of the variants that have been observed will be chosen for onward transmission are considerably less well understood. This choice is generally influenced by individual's own interests and biases such as a desire to identify with a social

**Funding:** AP was funded by a Heriot-Watt PhD Fellowship. The funders had no role in study design, data collection and analysis, decision to publish, or preparation of the manuscript.

**Competing interests:** NO authors have competing interests.

group, and from values that emanate from social, educational and economic institutions [24, 25]. Anecdotally, influences of the social context are observed in, e.g., individuals who produce colloquial language learned from friends among friends and individually learned less-than-perfect table manners when alone, but who produce impeccable language and table manners (learned from parents) in the presence of their children. Or who express their friends' views to their peers, and their teachers' views to their students. However, this question still requires empirical testing. This paper will use an experimental approach complemented by a computer simulation in the first study (to our knowledge) that addresses how selecting a cultural variant to transmit onwards is shaped by associative learning and context-congruence effects.

## 1.1 A new context-based transmission bias

Human cultural transmission is highly biased [26]. Multiple studies have revealed and examined transmission biases and the strength of their effects on which variants individuals adopt and produce [12, 27–30]. Boyd and Richerson [15] distinguished three types of biases: content-based or direct bias, model-based bias and frequency-based biases (Fig 1a–1c), all of which involve a (biased) evaluation of different observed variants by a learner. The present study proposes and tests a different type of bias which does not presuppose evaluation on the part of learners or a preference for specific variants. Instead, it entails a simple conditioning effect based on congruence, or similarity, of the current context in which a variant is produced and the context in which it was observed and learned (Fig 1d).

The role of context on learning is the focus of studies of transfer (see e.g. [31, 32] for reviews). There is transfer when something that is learned in a context or a domain is easily transferred or generalised to other domains. For example, when information that is learned in a class context is subsequently produced in a test context [33]. Transfer encompasses the assumption that what is learned in a social context carries over to other social contexts. This

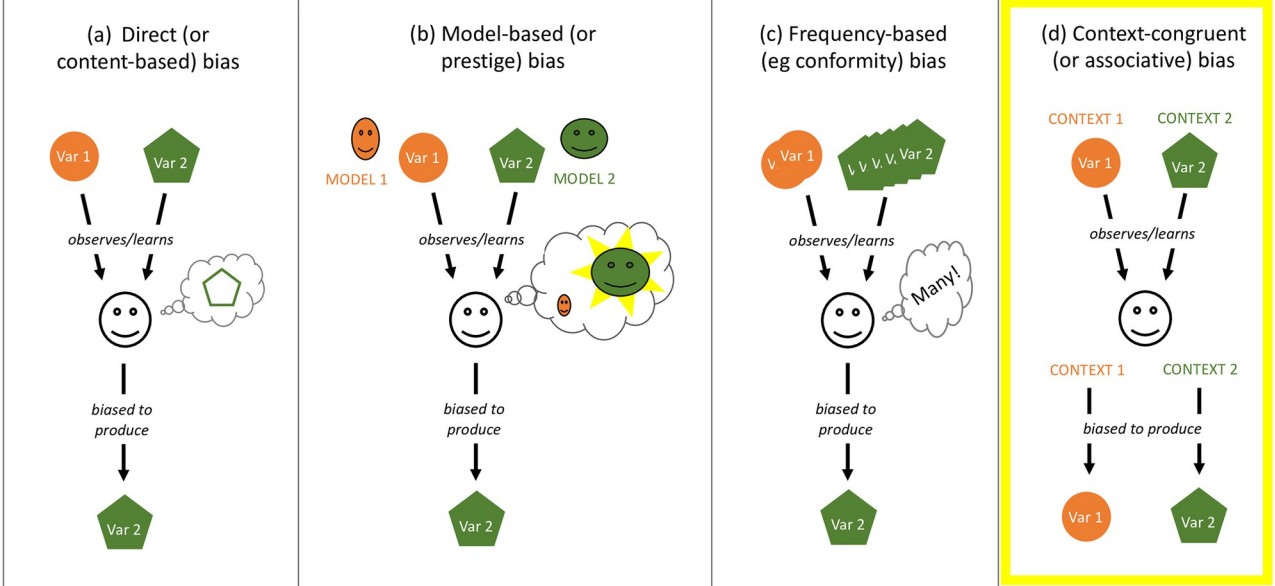

**Fig 1. Three classic transmission biases and a new one.** (a) Direct or content-based bias favours the adoption of variants depending on their perceived attractiveness, utility, ease etc. (b) Model-based bias favours variants depending on who produced (or modeled) those variants. (c) Frequency-based bias disproportionately favours variants that have high (or low) frequency. (d) Context-congruence or associative bias favours variants that are associated with the current context, i.e. that were learned, observed or produced in the same context).

assumption, however, has been heavily criticised [31, 34–37]. Transfer to new contexts seems to be the exception rather than the rule [34] and we learn separately how to act differently in different social and physical contexts [38], as the following examples illustrate. Displays of affection or emotion that are acceptable at home are not produced outside the home [39]. Babies behave in a depressed way with their depressed mothers, but behave normally when interacting with other caregivers [40]. Six-month old infants trained to kick their legs when they saw a mobile toy, did not kick in response to the mobile when incidental aspects of the context changed slightly (e.g., if the covering of the playpen was replaced with another of a different colour) [41]. Context-specificity can modulate not just behaviour, but cognitive skill (see [42] for review). These results suggest that what is learned in one context remains largely circumscribed to that context and emphasise the association between the information learned and the context in which it is learned.

The current study will explore the effects of congruence regarding a particular aspect of the contexts in which variants are learned and subsequently transmitted, namely the relationship between the model and the learner. The role of model-based bias on transmission [43] is therefore of relevance. This bias relates to how the choice of variant to be adopted by a learner is influenced by characteristics of the model, or transmitter, such as status [44], age [45, 46], knowledge [43], success [47] or similarity of the model to the observer (homophily) [48]. Henrich and Gil-White [43] concluded that individuals are biased to copy successful individuals who have real or perceived skill, a strategy that can prove adaptive, as those individuals will, potentially, be more successful than others in the same environment. Learners use social prestige and age as cues to infer models' expertise [43, 49]. For example, adults prefer to copy prestigious individuals (those who others spend more time observing [50] and children are more likely to copy other children when their actions are effective [51], but tend to copy adults over children [52, 53] regardless of whether they report being experts or not [54].

## 1.2 Relationship between model and learner: Transmission modes

Relationships between model and learner, related to model-based bias, are also connected with the cultural transmission modes defined by Cavalli-Sforza and Feldman [55]: vertical transmission, from a member of one generation to a biologically related member of a subsequent generation; oblique transmission, from a member of one generation to a biologically unrelated member of a subsequent generation; and horizontal transmission, between two members of the same generation.

Many cultural evolutionary studies conflate vertical and oblique transmission under the 'vertical' label (e.g., [56–58]) and use this to refer to transmission of information that that can persist over many generations as it passes from parents and experts to children and naïve individuals. In this study, we follow the same convention and focus on the contrast between transmission from experts and transmission among peers. We acknowledge that reverse or 'upward' vertical transmission from a young expert to an old novice can occur. However, this is only briefly mentioned in theoretical work, e.g., [59, 60], in contrast with an overwhelming focus on the downward pathway, e.g., [15, 55, 59, 61]. Even when cases of reverse cultural transmission are reported, they are treated as the exception rather than the rule, e.g., [62–65]. We contend that there is good reason to relate expert-to-novice transmission to vertical transmission.

Upward novice-to-expert (and upward child-to-parent) transmission occurs when an innovation emerges among or is accessed predominantly by young individuals and is initially not accessed or displayed to the same extent by older individuals. E.g., a young academic teaches a novel statistical method they learned recently at university, which is more effective or efficient

than older methods, to an older colleague. Other modern-day examples of cultural traits that are transmitted upwards include digital-native abilities, recycling behaviour, patterns of social media use or thumb-typing on a smartphone.

Whilst young individuals do transmit information to older ones, they will usually also transmit the same information to younger individuals, often to a greater extent. The young academic who passes on the new, better statistical method to older colleagues are likely to transmit it as well to a greater number of younger students. The few cultural trait variants that are transmitted upwards to a greater extent than downwards are culturally unstable, i.e., do not persist unchanged for long. Examples of such variants include the behaviour of an individual towards their poorly elderly parent that is not be witnessed by the individual's children; a pedagogical technique devised to 'teach new tricks to old dogs' employed by younger academics when they teach a new statistical method to older colleagues; and arguments regarding the use of HRT in menopause a woman tailors to persuading her mother and aunts (but not her daughters).

In brief, upward transmission of cultural traits, including expert-to-novice transmission, is rarely greater than downward transmission. Consequently, net transmission between experts and novices flows from older to younger individuals. For this reason, while acknowledging that there are many exceptions, in addition to associating peer-to-peer transmission with horizontal transmission, in this paper we associate expert-to-novice transmission with (classic, predominantly downward) vertical transmission.

A bias for vertical transmission (comprising vertical and oblique) is evidenced for many cultural traits in both children and adults. Children are more inclined to copy the actions of adults than those of other children [45, 66–68]. For instance, when witnessing the performance of novel actions by both adults and peers, children imitated the actions of the adult models over those of the child models [67]. Moreover, children more readily imitate the actions of a highly competent teacher model than those of a highly competent peer model [69]. Fourteen-month-old infants' imitative tendencies increased as the age of the model increased; imitating adults more often than younger models [67]. This leads to high degrees of concordance between parents and children's behaviour and attitudes [70–72]. Adults are also more likely to imitate the actions of those whom they perceived as their superiors [73]. Complex traits such as political tendencies [74–76], academic values [77], bidding behaviour in online lending platforms [78] as well as religion, entertainment, sports, superstitions and beliefs, customs and habits [79] tend to be transmitted vertically. Vertical transmission, mediated by factors including humans' capacity for faithful imitation [1, 10] and the socializing influence of educational and other institutions [25], allows the preservation of cultural traits not only from one generation to the next [55], but often for many generations [1, 3, 10, 20, 80].

A different set of cultural traits such as taste in clothes and hair [81], consumer socialisation [82], social skills [83], drinking behaviours [49, 84, 85], smoking [86, 87] and eating behaviours [88, 89] have been associated with horizontal transmission, a transmission mode that can only guarantee the conservation of information for one biological generation [55]. As children age, horizontal transmission becomes dominant [90], with learners becoming more likely to acquire the traits of their peers than those of adults [55]. Older infants retain significantly more information learned from peers than from adults than younger infants [66]. Horizontal transmission seems to be strongly associated with identity. Children copy more models who they identify with [67], even in the absence of communicative context, and identifying a model as being "like me" leads to the peer model advantage in infant imitation [91] (see also [92]). Children tend to copy children over adults in a novel toy task [45] but, generally, when ingroup identity is affected, they tend to imitate their peers' behaviours [93]. Rogers [94], argues that homophily, the tendency to imitate those who are similar to oneself, allows for

more efficient communication, and it is more likely to lead to behaviour change. The effects of a model-based "horizontal bias" in connection with identity or homophily have also been observed in adolescents and adults [95–97], with peers in these developmental stages deploying similar behaviours, in other words, showing behavioural congruence. When participants perceive a model as part of their group (similar to themselves), they are more likely to imitate their gestures than the gestures of a model whom they perceive as someone outside their group [98]. They are also more likely to imitate the gestures of a confederate when they perceived her to be a peer [73]. Finally, behaviour-change interventions delivered by research assistants (non-experts) had a larger impact on behaviour than when they were delivered by experts [99].

In sum, individuals acquire cultural information both vertically and horizontally. Learning from parents and experts is the default in younger and naïve individuals, while learning from peers is more likely when there is a perception of similarity between model and learner.

## 1.3 Associative learning

Associative learning takes place when a learner connects two events, where one event refers to, signals, co-occurs with, or causes the other [100–102]. It has been suggested that associative learning could account for imitation, the reproduction by a learner of behaviour observed in a model [103], and be involved in the acquisition of complex traits, such as word-learning [104] or social value [105]. Heyes and Pearce [106] have argued that associative mechanisms "make learning selective" (p. 6) and can be viewed as learning strategies. Thus, a learner may use social (and asocial) cues to "decide" which traits to acquire. During cultural transmission, specifically, learners could associate the variant they deem best to acquire for themselves with the transmitter who has a specific relationship with them. For example, an individual may acquire the political orientations of their parent(s) [74, 75] because of their parent-child relationship, while that same individual may acquire the social skills of a peer [83], because of their peer-peer relationship. Vertical cultural transmission may occur from just one parent to the child. There seems to be higher behavioural congruence in mother-daughter and father-son dyads than in father-daughter and mother-son dyads [107–109]. These patterns may be due to congruence between the contexts of learning and onward transmission, whereby a woman transmits to her daughter cultural variants she learned from her mother and a man transmits to his son variants learned from his father.

The factors influencing what learners transmit on once they become transmitters themselves are still not fully understood. When an individual has learned more than one cultural variant for a given function or problem, which variant will they choose to pass on? The current study explores the idea that congruence between the context of acquisition and the context of onward transmission will, by associative learning, influence this choice. We had participants learn two variant strategies from an expert and a peer and then asked them to transmit either to a naive participant or to another peer, in order to test the effects of congruence bias and its interaction with model-based bias.

We therefore hypothesise:

H1. Following the model-based bias literature, we expect that the variant learned from an expert will be produced more often than the variant learned from a peer.

H2. We predict an effect of context congruence according to which the production of the variant that was learned in a context that matches the current context is favoured. But this context-congruence bias will interact with the above-mentioned model-based bias. Assuming, for simplicity's sake at this point, that congruence bias is as strong as model-based bias, we formulate the following sub-hypotheses:

H2a. In the expert-to-novice production context, since model-based bias and congruence bias act in the same direction, the production of the expert's variant will be higher than the production of the peer's variant.

H2b. In the peer-to-peer context, model-based bias and congruence bias pull in opposite directions, the former favouring production of the expert's variant, and the latter favouring production of the peer's variant. In this condition, the two biases will cancel each other out, therefore we hypothesise that the peer's variant will be produced as often as the expert's variant.

However, we have no solid grounds to assume that both biases will be of equal magnitude, or that one will be stronger than the other. Therefore, in addition to testing the above hypotheses experimentally, we conduct an exploratory computer-simulation-based parameter estimation to measure the relative strength of model-based bias and congruence bias in our experimental results.

## 2 Methods

This study was granted ethical approval from Heriot-Watt University's School of Social Sciences Ethics Committee, and it was pre-registered with the Open Science Foundation (https://osf.io/6q4bn).

### 2.1 Participants

Sixty-four adults (33 female, M = 21.17 years, SD = 2.19 years; range = 18–28 years) were recruited from the student population at Heriot-Watt University (UK). Sixty-four was the minimum participant sample size stated in the pre-registration. We decided to run that number of participants and only examined the results after we reached it. Participants were recruited by the experimenter approaching them directly in a University social space or via an advert posted around the campus. These participants were entered into a raffle to win one of two Amazon vouchers (£25 each), with psychology undergraduates also receiving course credit for their participation.

In addition, 5 undergraduate students, all female and aged between 19 and 21 years (two studying second year, one third year, and two final year), acted as confederates in the experiment.

### 2.2 Materials

Using geargenerator.com we created a linear sequence of seven interconnected gears (Fig 2). The gears were presented on a computer display (sequence frozen at first) and participants were given the task to predict the direction (clockwise or anti-clockwise) in which the last gear in the sequence (coloured in blue for salience) would turn, if the first gear turned clockwise.

This task has been used previously in studies exploring problem solving in children [110, 111] and adults [112, 113]. It was chosen as it is readily understandable and has two alternative solutions (the two cultural variants learned and transmitted in our study) that are roughly equivalent in difficulty and take about the same time to complete, namely the 'parity' solution and the 'skipping' strategy. These two strategies were described to the participants as follows:

- Strategy A: Parity. "First, you count all the gears. If we have an even number of gears, then the last gear will turn in the opposite direction from the first. If we have an odd number of gears, then the last gear and the first gear turn in the same direction".

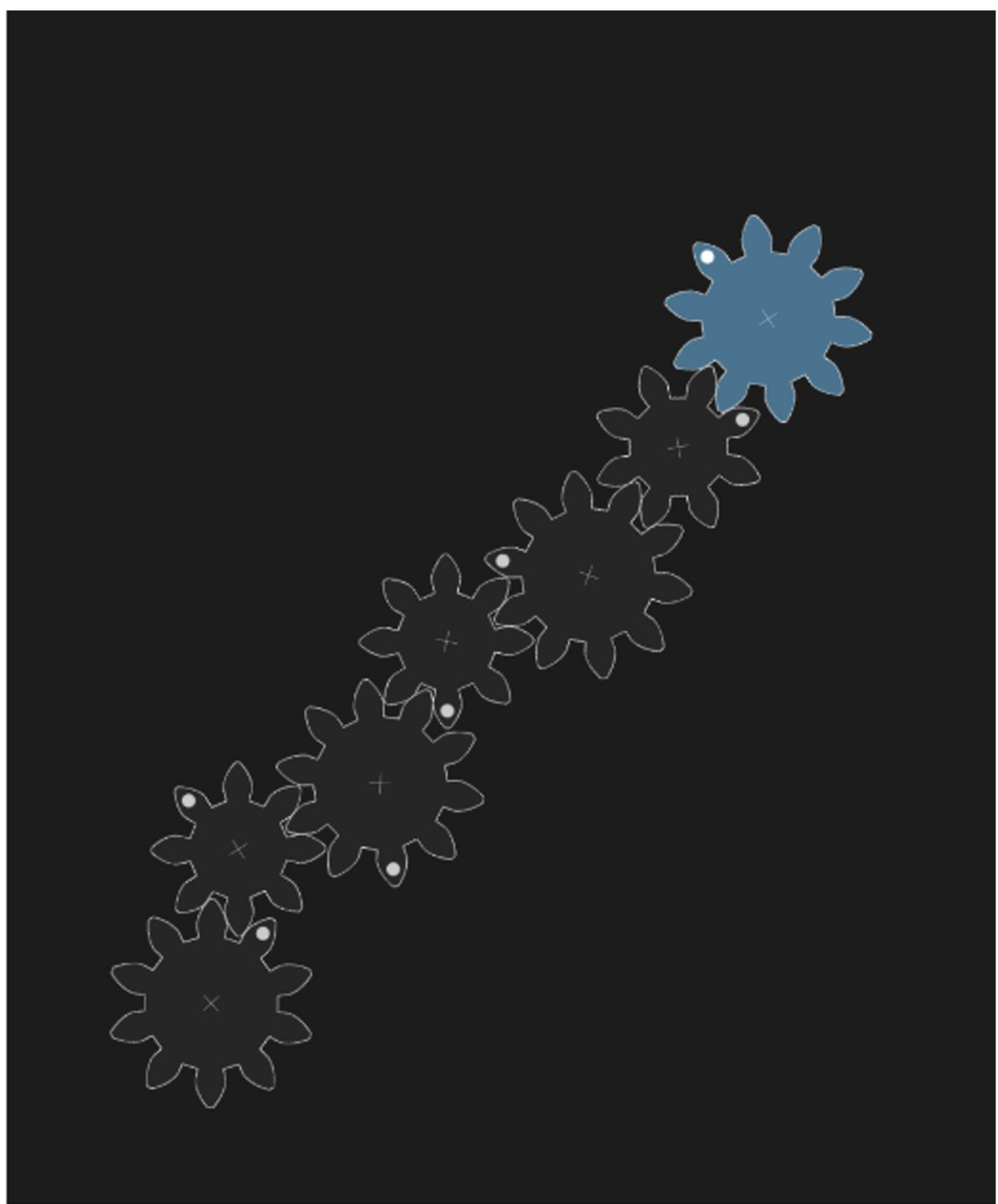

**Fig 2.**

| 1. Learning | | 2. Onward Transmission |
|---|---|---|
| Expert-to-novice context *P learns strategy A from expert* | *AND* Peer-to-peer context *P learns strategy B from peer* | Expert-to-novice context *P teaches a strategy (A or B) to a novice* OR Peer-to-peer context *P shows a strategy (A or B) to a peer* |

**Fig 3. Experimental design showing what a participant *P* learns and transmits onwards.**

- Strategy B: Skipping. [Model points to each consecutive gear in turn] "You can go 'clockwise, anticlockwise, clockwise, anticlockwise, clockwise. . .'. When you point at the last gear, you will be saying whether it will turn clockwise or anticlockwise".

The skipping strategy is usually discovered first by naïve individuals [110], and the parity strategy is rarely discovered by children [112], but once known they are roughly equivalent in difficulty and take about the same time to complete. We will check any bias for one or the other in our result.

## 2.3 Design

Throughout this paper we use the verb "teach" to describe instances where an expert transmits to a novice, and "show" for instances where a peer transmits to another peer.

The study employed a between groups design in which participants were randomly allocated to one of two conditions (Expert-to-novice or Peer-to-peer condition).

Each participant first learned two strategies to solve the gear problem in two contexts, one from an expert and one from a peer. The order of presentation of the contexts and strategies was fully counterbalanced. Then, half of the participants were in the Expert-to-novice condition, in which they were asked to teach a novice how to solve the problem. The other half were in the peer-to-peer condition, in which they were asked to show another peer how to solve it. The 'expert' was the experimenter and the 'novice' and 'peers' were three different confederates. Thirty-two participants were tested in each condition. The experimental design is summarised in Fig 3.

The dependent variable was the strategy produced in the context of onward transmission, either the strategy that was taught to the participant by a novice (in the Expert-to-novice context of learning) or the one that was shown to the participant by a peer (in the Peer-to-peer context of learning). The order of presentation of the two variant strategies (parity or skipping), and the two contexts of learning was fully counterbalanced (see Fig. SM_1 in S1 File).

## 2.4 Procedure

The experimenter welcomed the participant and led them to a waiting area, where they were introduced to the first confederate (C1), who pretended to be another participant (a 'peer' of the participant). The experimenter explained that the experiment would be conducted in pairs and led both the participant and C1 to the testing lab. On entry to the lab, both the participant and C1 were given a participant information sheet to read and a consent form to sign. Next,

they were given the following instructions: "I will present you with a problem and then I will teach you the solution. If you look at the computer screen, you'll see some gears connected to each other and (experimenter hits 'play') move each other. Now see, the first gear turns clockwise (experimenter points at the gear on the left top corner; then hits 'stop'). The problem is to figure out which way the last gear of the sequence turns". After that, one of four different dialogues occurred, according to the condition in which the participant was assigned (see SM, section 2 for all dialogues). Below is an example of one of the dialogues and actions for the condition in which the expert's strategy is taught first (Expert-to novice learning context first, Peer-to-novice second), and the participant then teaches a novice (onward transmission context is Expert-to-novice):

- Experimenter: "I will teach you the solution I've taught many people before as part of my experiment." [Teaches strategy].

- C1 (Peer): "I've played this before, I know another solution." [Shows alternative strategy].

- Experimenter—[Initiates Phase 2]: "Oh okay, that's interesting! So now you both know two different solutions to this problem. Okay, for the next part I need you both to teach your solution to two other participants who don't know how to solve the problem yet. So, I'll need one of you in this lab and the other one to the other lab. Who would like to come with me to the other lab?"

- C1 (Peer): "I'll come."

- [The experimenter leads C1 out and, after approximately ten seconds, brings in a different confederate (C2): the novice.]

- Experimenter: [looks at C2] "Okay so, the problem is to figure out which way the last gear turns. . . [Participant's name] is now an expert at this, and he/she will now teach you his/her solution to the problem."

At this point, the experimenter left the testing lab. Both she and the first confederate were absent during the onward transmission phase (from participant to a different confederate) to remove demonstrator presence-related pressures [4, 114, 115]. After approximately thirty seconds (and after listening through the door to make sure that the participant had finished teaching/showing their solution) the experimenter returned to the testing room and asked C2 to go back to the waiting area (from where she supposedly brought them) to await debriefing. The confederate then left the testing room and noted the solution that the participant chose to pass on (so that they could inform the experimenter after the participant left). Next, the experimenter asked the participant a series of questions that aimed to determine why a particular solution was transmitted and whether the participant suspected that C1 and/or C2 were confederates: (1) Why did you teach/show that solution? (2) What did you think happened in this experiment? and (3) What did you think of the people who were with you in this experiment? Finally, the experimenter debriefed the participants and thanked them for their participation.

### 2.5 Coding and analysis

For each participant, we coded their anonymised ID, gender and age, and 5 variables: three control factors (strategy produced, Parity or Skipping; order of strategy, Parity first or Skipping first; and order of acquisition context, Expert-to-novice first or Peer-to-peer first); one independent factor, the context of onward transmission (Expert-to-Novice or Peer-to-Peer).

## 3 Analysis and results

### 3.1 Hypothesis testing and control conditions

Two male participants (both from the Expert-to-novice condition) were excluded from the analysis, as the post-test questionnaire revealed that they suspected the presence of confederates. No other participants indicated that they suspected this during the post-test questions. The data and the analysis are in github.com/mtamariz/ContextCongruence/. We used $X^2$ to test our three hypotheses with a Bonferroni-corrected α = 0.0167.

We hypothesised an effect of model-based bias whereby the Expert's variant would be favoured overall (hypothesis H1). Indeed, the Expert's variant was produced more often (41 times) than the Peer's variant (21 times), and this difference is significant ($X^2$ (1) = 6.45, $p$ = 0.011).

We also hypothesised an interaction between context-congruence bias and model-based bias. When transmitting onwards in the Expert-to-novice context, participants produced the Expert's variant 27 times and the Peer's variant 3 times (Fig 4), a significant difference ($X^2$ (1) = 19.20, $p$ < 0.001), in support of Hypothesis 2a. In the Peer-to-peer context, participants produced the (congruent) Peer's variant 18 times and the (incongruent) Expert's variant 14 times (Fig 4), which is not significantly different ($X^2$ (1) = 0.50, $p$ = 0.480) in support of Hypothesis 2b.

Regarding control variables, neither variant strategy, order of learning nor order of presentation had no effect on production. The parity variant strategy was produced 34 times, and the skipping variant, 28 times, a non-significant difference ($X^2$ (1) = 0.58, $p$ = 0.45). Further chi-

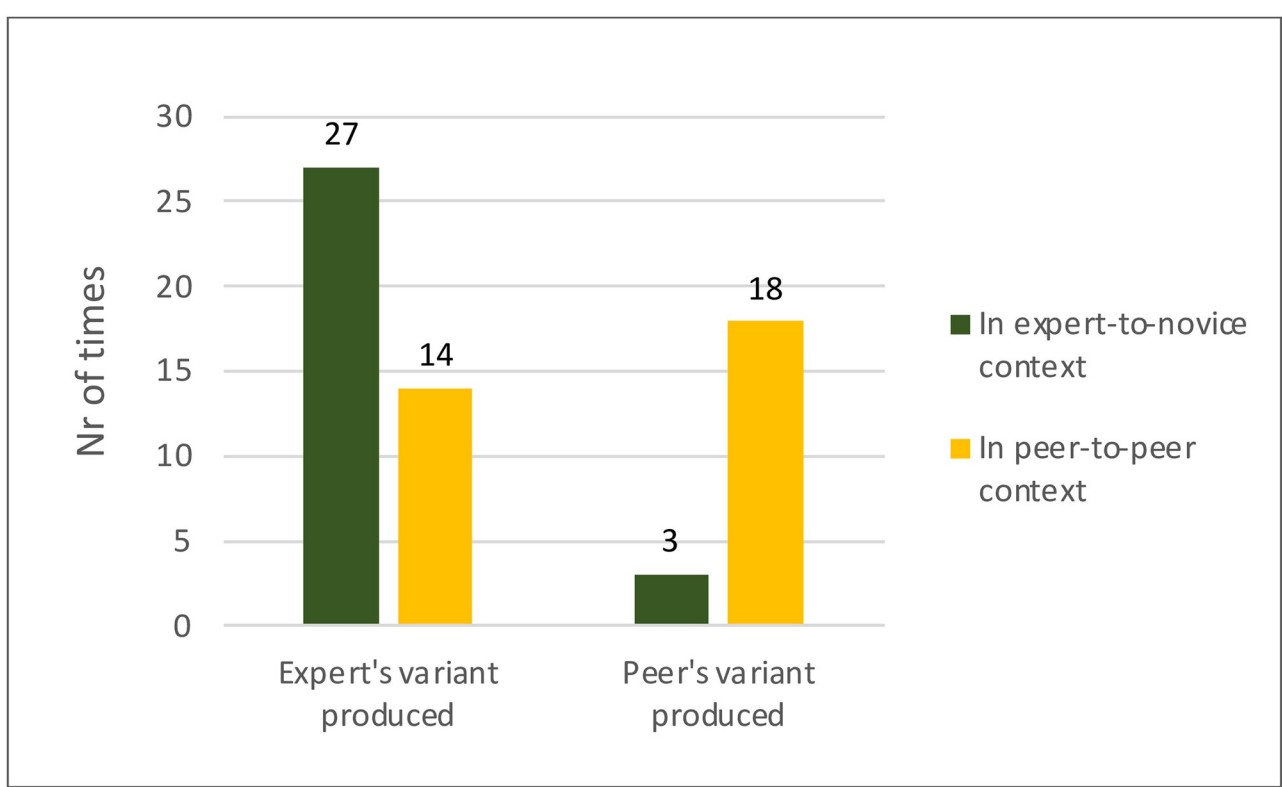

**Fig 4. Experimental results showing the number of times the expert's and peer's variant strategy was produced in each onward transmission context.**

squared tests of independence indicated that the transmitted variant (Expert's or Peer's) was neither associated with the order of learning contexts (first from an Expert or first from a Peer ($X^2$ (1) = 0.52, $p$ = 0.47), nor with the order of presentation of the Parity and Skipping variants, ($X^2$ (1) = 0.13, $p$ = 0.72).

These results are compatible with the presence of two simultaneous biases on the choice of variant to transmit onwards: model-based bias favouring the production of the variant learned from the expert, and context-congruence bias favouring the variant learned in the context that matched the current production context. Fig 4 gives us an impressionistic idea of the relative strength of these two biases. The combined action of the two biases yields a large advantage of the production of the expert's variant in the Expert-to-novice context, while their opposed action yields no significant difference in the Peer-to-peer context.

These analyses, however, do not quantify the strength of the biases. In order to obtain those strengths, we constructed computer simulations of the experiment to estimate the biases' magnitudes.

### 3.2 Parameter estimation simulation

Our experimental results were not consistent with the null hypothesis predicting that the Expert's and the Peer's variant would be produced with equal probability, regardless of the context of transmission. Instead, these results may be explained by three forces, operating alone or in combination: Model-based or prestige bias, context-congruence bias and order effects (primacy bias favouring the production of the variant that was learned first, and recency bias favouring the variant that was learned last). To test the extent to which they affected the participants' choices we had pre-registered an analysis based on generalised linear mixed models. Given that GLMER models often did not converge in part due to small sample sizes, and the impracticality of collecting more data after this was discovered, we took instead a Monte Carlo simulation approach [116]. We constructed computer simulations of the experiment including parameters representing the three forces. We ran the many simulations with a large sample of combinations of parameter values and counted how many of the simulation run results matched the veridical experimental results. We inferred that the parameter value combinations that best fitted the experimental data reflected the strength of the biases shaping the participants' choices.

### 3.3 Estimating the magnitude of expert and congruence biases

We constructed a simulation to estimate Expert and Congruence parameter values. (The code can be found in github.com/mtamariz/ContextCongruence/). The simulation (described in Table 1) looks for the combinations of Congruence and Expert bias values that best fit our experimental results. These parameters are defined as follows:

- *ExpertBias* is the probability that the variant learned from an expert is produced. It takes values between -1 and 1. Positive values indicate a preference for the Expert's variant and negative values indicate a preference for the Peer's variant. A value of 0 indicates no bias. *PeerBias*, or the probability that the variant learned from a peer is produced, equals, therefore, -*ExpertBias*).

- *CongruentBias* is the probability that the variant learned in the current context is produced. It takes values between -1 and 1, with positive values indicating a preference for the congruent variant, and negative values indicating a preference for the incongruent variant. A value of 0 indicates no bias. (*IncongruentBias*, or the probability that the variant learned in a context different from the current context is produced, equals, therefore, -*CongruentBias*)

**Table 1. Description of each step of the simulation.**

| Pseudo code | Explanation | Example |
|---|---|---|
| for *ExpertBias* (-1:1) | Select each value of *ExpertBias* between -1 and 1. | *ExpertBias* = 0.34 |
| for *CongruentBias* (-1:1) | And each value of *CongruentBias* between -1 and 1. | *CongruentBias* = 0.39 |
| for *Sim* (0:S) | Run S simulations, e.g., S = 1000. | *Sim* = 16 |
| for *Partic.* (1:P) | For each participant in the current experimental condition | P = 21 |
| Sample variant | Select a variant to produce according to parameter values in the current condition (see Table 2) | *Var* = E |
| Get *Counts* | Count number of participants that selected Expert's and Peer's variants in each condition in this *Sim* | *Counts* = (20, 11, 2, 23) |
| Check match | Check whether all four *Counts* obtained in this *Sim* equal experimental counts (which are 27, 18, 3, 14, see Fig 4). | 20 = 27? NO<br>11 = 18? NO<br>2 = 3? YES<br>23 = 14? NO<br>Match = FALSE |
| Get *Matches* | Count number *Sims* where simulated *Counts* match experimental counts | *Matches* = 16 |

We simulated our two experimental conditions: transmit a variant strategy in an Expert-to-novice context, and transmit a variant strategy in a Peer-to-peer context. Within each condition, one of these two variants was produced. Production could therefore be congruent (when the variant produced had been learned in the context matching the production context) or incongruent (when it had been learned in the non-matching context). In each case, production of variant strategies was influenced by *ExpertBias* and *CongruentBias* as shown in Table 2.

Each simulation of our experiment returned the number of times the 62 participants had produced the Expert's and the Peer's variants in each condition. Since the parameter values only affect production probabilistically, each simulation returned different numbers. The veridical values obtained in our experiment in each condition are those in Fig 4. For each parameter combination, we counted how many times the simulation values matched the experimental values, an estimation of how likely it is that those parameter values represent the strength of the biases guiding our participants' choices. As proxies for the uncertainty of the parameter estimates we calculated the standard deviations of the distributions of matches and show the colour gradient visualisation (Fig 4).

We used t-tests to estimate deviations from null hypotheses. Note these t-tests were not applied to simulated data, which can obtain arbitrarily high p-values by increasing sample sizes [117], but to distributions of *matches* between simulated and experimental data, which do not suffer from that problem.

Fig 5 shows the results of the simulations. The distribution of matches for *ExpertBias* had a significantly positive mean ($M = 0.433$, $SD = 0.1130$; one-sample t-test: $t(7700) = 286.85$, $p < 0.001$) indicating a preference for the variant learned from an expert.

The distribution of matches between experimental and simulation results for *CongruentBias* also had a significant positive mean ($M = 0.530$, $SD = 0.118$; one-sample t-test: $t(7700) = 394.95$, $p < 0.001$) indicating a preference for the congruent variant.

**Table 2. Calculation of the probability that the congruent and incongruent variants are produced in each experimental condition as a function of *ExpertBias* and *CongruentBias*.**

| | Condition: Context of onward transmission | |
|---|---|---|
| | **Expert-to-novice** | **Peer-to-peer** |
| *P*(Congruent) | ExpertBias x CongruentBias | (1-ExpertBias) x CongruentBias |
| *P*(Incongruent) | ExpertBias x (1 − CongruentBias) | (1-ExpertBias) x (1 − CongruentBias) |

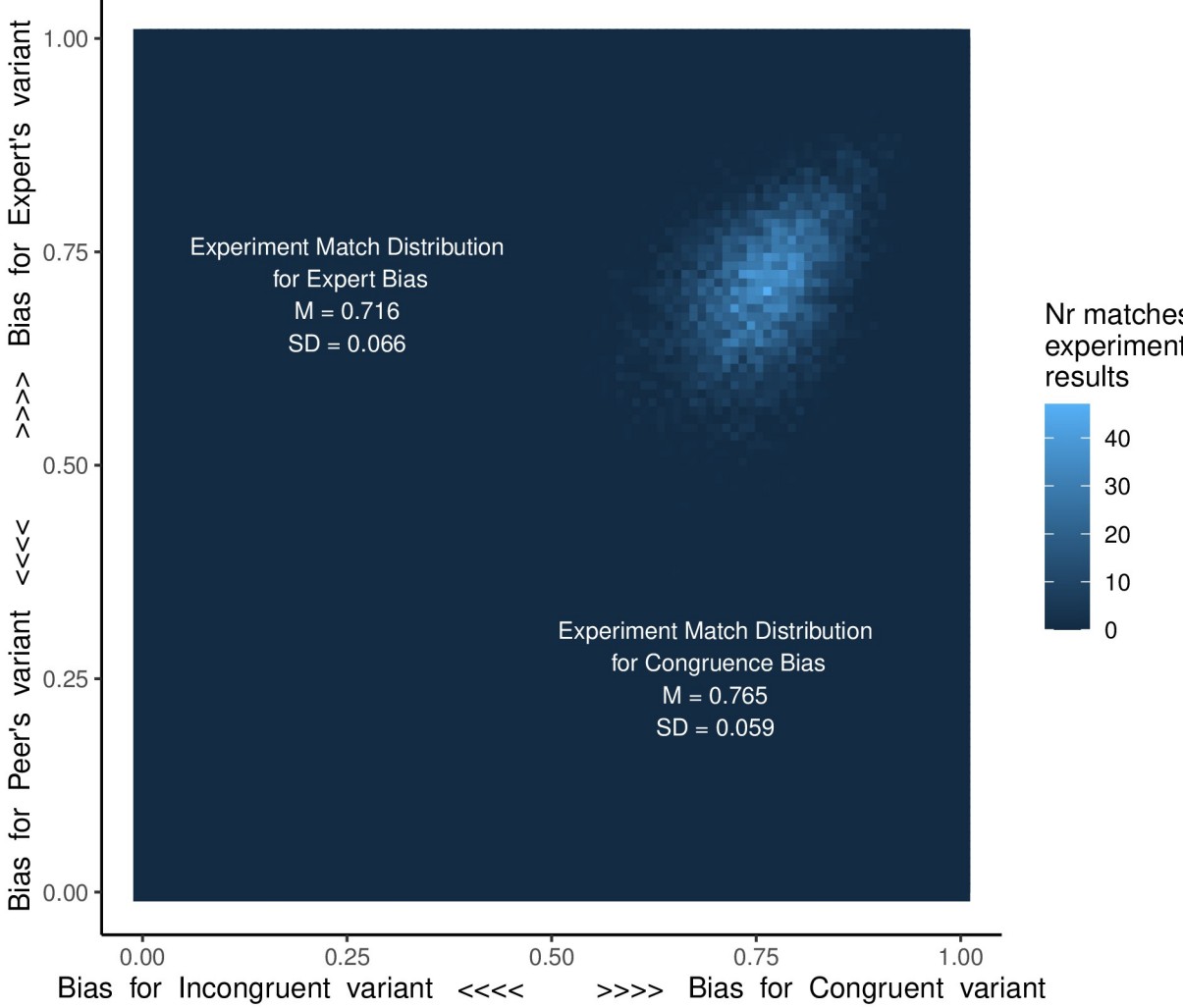

**Fig 5. The *CongruentBias* x *ExpertBias* parameter space showing the number of times (out of 5000) that the simulation results matched the experimental results for each parameter value combination.** Lighter colour represents more matches, indicating the parameter values that best fit the experimental results. Matches cluster in an area of around positive values of both *CongruentBias* and *ExpertBias*, indicating biases in favour of the congruent variant and the Expert's variant.

Those two distributions were significantly different (two-sample t-test: $t(15195) = 48.241$, p < 0.001). The effects of *CongruentBias* were, therefore, stronger than the effects of *ExpertBias* in our experiment.

We ran with simulations in all the combinations of *ExpertBias* values between -1 (strongest bias for the Peer's variant) and 1 (strongest bias for the Expert's variant), by increments of 0.01; and *CongruentBias* values between -1 (strongest bias for the incongruent variant) to 1 (strongest bias for the congruent variant), by increments of 0.01. Each combination of parameter values was run 5000 times.

### 3.4 Adding order effects

Aside from a preference for variants learned form an expert and for congruent variants, our results might be explained by an order bias (*PrimacyBias*: preference for the first variant (Parity or Skipping) learned; *RecencyBias*: preference for the last variant learned). To investigate

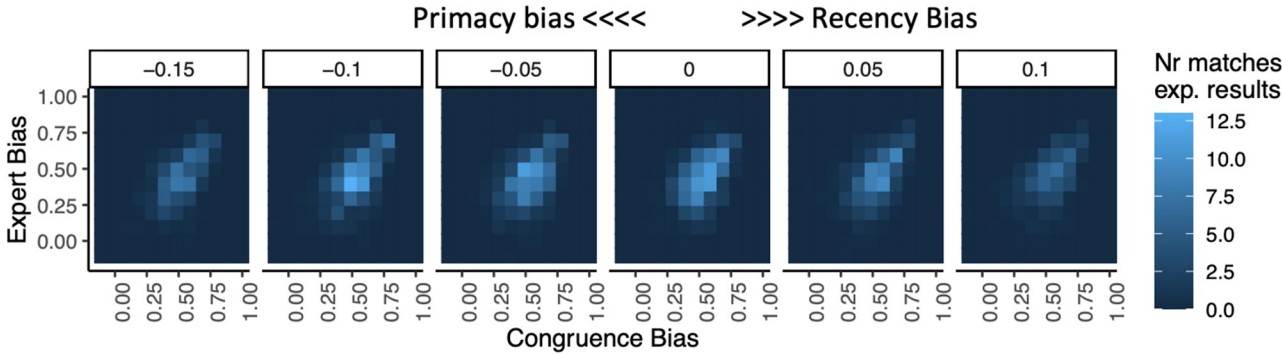

**Fig 6.** *CongruentBias* and *ExpertBias* parameter spaces for different values of *PrimacyBias*, showing the number of times (out of 5000) that the simulation results (approximately) matched the experimental results, for each parameter value combination. Lighter colour represents more matches, indicating the parameter values that best fit the experimental results. Only positive values of *CongruenceBias* and *ExpertBias* are shown, as there were no matches in negative values (i.e., values favouring the incongruent and peer's variants, respectively).

this possibility, we extended the simulation (The code can be found in github.com/mtamariz/ContextCongruence/) to include a new parameter:

- *PrimacyBias* determines the probability that a variant (parity or skipping) is produced, depending on whether it was learned first or last. This parameter takes values between -1 and 1. Positive values indicate a preference for the last variant learned; negative values indicate a preference for the first variant learned. A value of 0 indicates no bias. *RecencyBias*, or the probability that the variant that was seen last is produced is, therefore, (-*PrimacyBias*).

Production of variants in this simulation is affected by the biases shown in Table 2 and additionally include the *PrimacyBias* parameter (Table SM_1 in S1 File). We ran simulations with all the combinations of *ExpertBias* values between -1 (strongest bias for the Peer's variant) and 1 (strongest bias for the Expert's variant), by increments of 0.1; *CongruentBias* values between -1 (strongest bias for the Incongruent variant) to 1 (strongest bias for the Congruent variant), by increments of 0.1; and *PrimacyBias* values between -1 (strongest bias for the First variant seen) and 1 (strongest bias for the Last variant seen) in increments of 0.05. Each combination of parameter values was run 5000 times. Unlike the simulations above (Table 1), this time simulated results that were equal to the experimental counts, plus or minus one, were counted as matches. This was done because the probability of simultaneously finding 8 identical counts (see the 8 experimental counts in Table SM_1 in S1 File) in the same simulation was vanishingly small.

Fig 6 shows counts of simulation runs that match the experimental data for values of *PrimacyBias* close to 0 (the neutral point where neither primacy nor recency effects are at work). Matches to experimental data are found around the same values of *ExpertBias* and *CongruentBias* for different values of *PrimacyBias*. The distribution of *PrimacyBias* values matching the experimental results ($M = -0.037$, $SD = 0.125$), showed a small but significant preference (one-sample $t$-test: $t(536) = -6.77$, $p < 0.001$) for the first variant learned. (See the distribution of matching simulations over values of Primacy Bias in Fig. SM_2 in S1 File.)

## 4 Discussion

This study set out to test whether congruence between the contexts in which a cultural variant is learned and transmitted onwards affected its transmission, and to estimate the magnitude of this congruence bias relative to a well-attested model-based bias. In our experiment, two

models, an expert and a peer, each taught the participant a different strategy for solving a problem. The participant was then asked to transmit onward only one of the strategies, either to a novice or to another peer. The results show evidence for biases in favour of transmitting onwards the expert's variant, and also of transmitting onwards the variant that had been learned in the congruent context, which support our hypotheses. Furthermore, a series of simulations designed to find the bias values that best fitted the experimental results returned a Congruence bias of greater magnitude than the model-based bias for the Expert's variant. A small but significant Primacy bias in favour of the first variant learned was also found.

Regarding the magnitude of the biases, Congruence bias has a value of 0.53, intermediate between 0 (no bias) and 1 (maximum preference for the expert's variant). Similarly, Expert bias has a value of 0.433, also intermediate between 0 (no bias) and 1 (maximum preference for the expert's variant). Primacy bias has a value of -.037, much closer to 0 (no bias) than to -1 (maximum preference for the first variant learned). Without more experimental evidence, we cannot be certain of whether biases increase linearly or not. Congruence bias is stronger than Expert bias, and both seem to be much stronger than Primacy bias in our data, but in order to fully understand these relationships, more studies are needed.

## 4.1 Model-based, vertical or expert bias

In line with previous studies [44, 45, 50, 66, 68] transmission was affected by a model-based vertical bias, with the expert's variant strategy being more likely to be transmitted onward than the peer's. Generally, model-based biases [15] (see also [118]) posit that copying one variant over another is affected by attributes of the model/transmitter who exhibits that variant [5, 53, 119, 120]. In particular, copying an expert may be due to perceived skill and knowledge [43, 47]. The experimenter (the expert in our experiment) explicitly displayed these attributes, by presenting herself as a "teacher" who has "taught many people before" in "her" experiment. In contrast, the peer model (confederate 1) and the learner (confederate 2) were presented as naive participants about to learn something and "a participant just like" the participants themselves, reinforcing the perceived homophily and equality in skill and knowledge.

## 4.2 Associative learning and context-congruence bias

However, despite the preference for the expert's variant, our findings show that learners do not copy experts unconditionally. When transmitting to a novice (in an expert-to-novice context) the expert's variant was overwhelmingly preferred. But when transmitting to a peer, the peer's strategy was actually produced slightly more often than the expert's indicating an advantage of the congruence context bias over the expert bias. Our simulation confirmed a stronger bias in favour of the variants learned in the congruent context ($M = 0.530$) than in favour of the Expert's variants ($M = 0.433$).

This is strong evidence for a context-congruence bias operating on transmission for a particular aspect of the context, namely the social relationship between model and learner. The relationship was constructed in the experiment by telling participants either that they were "now an expert", or that they had to "show a peer" how to solve the puzzle. This intervention construed the transmission context as similar to one of the learning contexts. The fact that this was the only difference between the two experimental conditions suggests that the context-congruence effect we found is due to associative learning [100–103, 106].

Although previous studies have also demonstrated the intergenerational congruence of traits during vertical (e.g., parent-to-child, [95–97]), and horizontal transmission (e.g., peer-peer-to-peer, [121, 122], the cognitive processes in operation leading to that congruence had not been sufficiently explored. This might be due to the narrower scope of focus of social

transmission studies, which address questions such as how the behaviour of peers affects the behaviour of participants [122], or how the closeness of a parent and child affects their behavioural congruence [123–126]. In other words, they were limited to the effects of the context of learning. The current study widened that scope by examining how relationships between the contexts of learning and onward transmission affect the social transmission of cultural traits. This illustrates the important role of associative learning in social transmission and the resulting congruence between transmitted behaviour in acquisition and transmitted behaviour in onward transmission.

## 4.3 Primacy bias

It is difficult to discuss the small primacy bias we found in our experiment, as few studies explore order effects on imitation or adoption of cultural variants, and the scant results are conflicting. Participants asked to imitate action sequences show a strong primacy effect and a weak recency effect (participants copied more faithfully the initial actions than the final ones, but both initial and final were copied better than the actions in the middle) [127]. But participants given opposed moral arguments, a recency effect on adoption was observed [128, 129]. Perhaps more relevantly, it has been hypothesised that the first piece of information observed or produced regarding a new topic or skill becomes an "anchoring hypothesis" that strongly affects subsequent behaviour and conclusions, and is difficult to reverse by subsequent information [130, 131]. It would be interesting to explore whether the first variant learned becomes more strongly associated with the task than subsequent variants during learning, or whether this bias is elicited during remembering for production.

## 4.4 Cultural-evolutionary consequences of congruence effects for vertical and horizontal transmission contexts

Our results provide, for the first time, direct evidence for a factor affecting not the adoption of behaviour by a learner, but specifically the onward social transmission of behaviour by learner: we tend to transmit to novices what we learned from experts and to peers what we learned to peers. This context-congruence effect has important implications for cultural evolution, as it can help us understand the mechanisms behind the persistence of vertically transmitted [79] cultural traits, and the lower continuity and faster change of horizontally transmitted traits [132].

Context-congruence biases entrench the reliance of cultural variants on a particular transmission pathway. Certain cultural traits including ideologies and values tend to be passed on from parents to children [70–72, 77, 79]; context-congruence bias means that learners will then transmit these traits to their own children, even if in the intervening time they acquire different orientations from peers. In this way, the variants will continue to follow that same vertical pathway over generations [10, 20, 133].

Similarly, context congruence predicts that predominantly horizontally transmitted cultural variants will tend to continue to follow this transmission mode. Horizontal transmission can lead to vast cultural change [3, 10, 20, 55, 80, 133], especially in the Information Age. Context-congruence bias could be behind the increasing peer-to-peer transmission of traits through social media and could therefore help explain the spread of maladaptive traits such as fake news and disinformation among online cohorts [134–136]. Understanding this bias could therefore be crucial for identifying ways of tackling problems arising from such phenomena, such as the aggravation of disease outbreaks [137, 138] or flourishing of racist attitudes [136, 139].

Theoretically, in cases of strict, exclusive horizontal transmission, cultural variants would be learned only among age-peers, become locked in a particular generation and die out with that generation (an example approximating this is the case of slang words that characterise a generation [140]). In contrast, variants that are transmitted purely vertically may persist for a long time, over many generations. Understanding the patterns of transmission for different traits and the strength of these effects could be used to fit parameterised evolutionary models and help predict the longevity of cultural traits.

Context-congruence biases operating on transmission modes, such as in the current study, however, do not predict that cultural traits will be deterministically 'trapped' in a particular transmission mode. Just as context congruence interacts with model-based biases—as shown in our results—it can also interact with content-based, frequency-based or other biases. Context-congruence bias may operate as a cultural selection mechanism increasing the overall (cultural) adaptiveness of traits. Cultural variants learned horizontally tend to be transmitted onward horizontally, and therefore tend to be short-lived. But some of these variants may be favoured by other biased bias, such as content-based or frequency-based bias, so strongly that context-congruence bias is overturned. For instance, if a variant learned from a peer is very functional, beneficial, attractive, easy to transmit etc, it may be passed on not just to peers, but also to novices and children. Especially adaptive variants may 'escape' the limiting horizontal transmission mode and become vertically transmitted.

Our study emphasises the distinction between adoption and transmission. Studies exploring how transmission biases (e.g. content-, model- and frequency-based) guide the adoption of cultural variants implicitly assumed that adoption predicted transmission, in other words, that if an individual adopted variant A after observing and evaluating (under bias) variants A and B, they would also transmit onwards variant A. Or, conversely, that production of variant A was evidence of adoption of this variant. We show that this is not necessarily the case. In our experiment, model-based bias predicts that all participants adopt the expert's variant. (We did not test this directly: we did not ask participants to solve the problem in the absence of a learner). However, in the context of peer-to-peer transmission, participants transmitted the (assumed) non-adopted variant, the peer's variant. This distinction is an important one to consider in theories about the transmission and spread of cultural variants.

## 4.5 Further exploration and applications of context-congruence biases

This study has examined only one very specific contextual aspect that may recur across the contexts of learning and transmission, namely the knowledge-balance relationship between model and learner (expert-novice or peer-peer). It will be interesting to test to what extent context-congruence bias operates for further model-learner relationships, e.g., Do we transmit to strangers what we learned from strangers and to friends what we learned from friends? Do we pass on to females what we learned from females and to our men what we learned from men? Exploration of gender was precluded in our study because we controlled for it: all the transmitters during acquisition (i.e., the experimenter and the confederates), as well as the learners during onward transmission (confederates) were female. However, future investigations of gender-based congruence bias could widen our understanding of the cultural transmission and evolution of gender roles and identities. Similarly informative would be the exploration of effects of congruence based on e.g., race [141, 142] and social class [143]. An additional question to explore is whether those context-congruence effects affect different cultural traits differently, e.g., is health-related information transmitted preferentially over maternal lines and political information over paternal lines? Does gender-based context congruence affect ideas but not behaviours?

Beyond social relationships, multiple other dimensions of the context could potentially affect transmission including place, time of day, time of year, weather, language spoken (for multilingual individuals) among many others. Understanding the relevance, relative strength and interactions between congruence biases based on all these aspects would be invaluable to inform and focus behaviour-change interventions. Suppose, for example, that a child is taught to recycle at school, but her family does not recycle at home. Place-based context-congruence will bias them against recycling at home. Social-relationship congruence will bias against passing on recycling to their children. In order to promote recycling at home, then, a strong intervention, perhaps based on content, model or frequency will be needed. But just how strong? In the same way as our simulation estimated the relative magnitude of expert and congruence biases, exploring the strength of context-congruence bias for different contextual aspects—and also compared to other transmission biases—for different behaviours and ideas will help gauge the necessary strength of interventions to promote (or dampen) transmission of specific traits. It will also help evaluate whether interventions have worked. The magnitude of context-congruence bias is an indicator of which contextual aspects are most salient and therefore most relevant to cultural transmission. Thus, evidence of, for instance, stronger gender-based congruence bias in older people than in younger people tells us that gender is less salient for younger people, perhaps indicating an attenuation in gender discrimination.

Considering context-congruence bias may help refine the design of studies exploring the spread of information on social media and therefore improve our prediction or prevention of the transmission of e.g., public interest knowledge or misinformation, respectively.

## 4.6 Experimental design issues

We used a complex experimental design, involving the construal of three 'cultural generations' (the model, the participant and the final learner) through the use of confederates. For success, it was essential that participants believed that the confederates were co-participants in the experiment. The post-experiment questions revealed that two participants suspected that the confederates were, in fact, part of the experiment (their data was not included in analysis). Although none of the remaining participants declared having realised the presence of confederates, one could be completely certain that they associated with the transmitter-confederate to the point that they felt she was their peer. Her additional knowledge regarding the experiment may have hindered somewhat the participant's ability to see her as their peer. In real-world situations a learner acquires cultural information from a peer-transmitter who possesses knowledge that the learner does not. Yet, the transmitter will still be identifiable as a peer, albeit being more knowledgeable. But participants would also see themselves as also slightly more knowledgeable peers when transmitting to another peer. In our study, the difference of knowledge between the participants and the confederates did not preclude their perceiving the confederates as similar to themselves, or their associating the peer's strategy with the one deemed best to transmit onward to them. Thus, even with this potential limitation, we observed context congruence. In the case of real peers and real experts (or real parents, etc.) the effect may be stronger.

The participants' perception of their knowledge status in relation to their perception of both the transmitter and the learner's knowledge status was another essential tool in our design. When introduced to the "novice", participants were referred to as "experts" to facilitate the perception of their difference in knowledge/expertise. They were introduced to the (confederate) "peer" as "another participant just like you" to strengthen the perception of peer status. However, it is difficult to ascertain the dynamics of the participant-to-learner relationship during onward transmission, as we had no way of measuring the increase or decrease in

perceived expertise and/or peer status. People experience analogous situations in everyday life, when they feel like "experts" (e.g., when teaching a younger cousin how to win at a game or teaching a sibling how to solve their math problem etc.) or "peers" (e.g., when sharing a post online with friends, when giving a recipe to a friend etc.). In these situations, their perceived role as transmitters is not always explicitly mentioned (as in our experiment), but implied. Our results, nevertheless, suggest that our manipulation was sufficient to make them feel like experts or peers according to the experimental design.

## 5 Conclusion

In sum, we have provided evidence for a novel bias in cultural transmission that links learning and onward transmission and is mediated by associative learning. In our study, cultural variants were more likely to be passed on if aspects of the current context matched aspects of the context in which the variant was learned. Our participants learned a variant strategy to solve a puzzle from an expert and another one from a peer. When asked to transmit to a novice, they were more likely to transmit the expert's variant; when asked to transmit to a peer, the difference disappear. We simulated our experiment in a parameterized model and the best fit of the experimental results were obtained with a model-based Expert bias value of 0.433, indicating an intermediate preference for the expert's variant and a stronger context-congruence bias value of 0.530, indicating also an intermediate, but significantly stronger, preference for the variant that was learned in the congruent or matching context, in other words, the context with which the variant was associated at the time of learning.

Context-congruence bias may amplify the endurance of vertically transmitted cultural traits such as language or religion and further reduce the spread of horizontally transmitted traits such as fashion or musical tastes. Exploring this type of bias for different aspect of the context (e.g., gender, age, place, time) and for different cultural traits (e.g., political orientation, language, environment-protecting habits); and studying its interactions with other transmission biases (e.g., content-, model-, frequency-based) will reveal the interplay and relative influence of the multiple forces that shape cultural transmission. Additionally, the outcomes of this exploration will offer detailed information to guide behaviour change interventions. For all these reasons, we propose that context-congruence bias is a significant addition to cultural evolutionary theory.

## Supporting information

**S1 File.**
(DOCX)

## Acknowledgments

A.P. was supported by a PhD fellowship granted by the School of Social Sciences of Heriot-Watt University. We acknowledge the help and suggestions from the student who were the confederates in the experiment: Claire Rogers, Kayleigh Lamond, Monica Ghoyal, and Shana Faraghat.

## Author Contributions

**Conceptualization:** Aliki Papa, Mioara Cristea, Nicola McGuigan.

**Data curation:** Aliki Papa.

**Formal analysis:** Aliki Papa.

**Investigation:** Aliki Papa.

**Methodology:** Aliki Papa, Mioara Cristea.

**Project administration:** Monica Tamariz, Aliki Papa.

**Resources:** Monica Tamariz, Aliki Papa.

**Software:** Monica Tamariz, Aliki Papa.

**Supervision:** Mioara Cristea, Nicola McGuigan.

**Validation:** Monica Tamariz, Aliki Papa.

**Visualization:** Aliki Papa.

**Writing – original draft:** Monica Tamariz, Aliki Papa.

**Writing – review & editing:** Aliki Papa, Mioara Cristea, Nicola McGuigan.

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
