## [Decision Letter · Decision Letter 0]

25 Jul 2022

PONE-D-22-12845A new modulator of cultural transmission: congruence between learning and onward transmission contextsPLOS ONE

Dear author,

Thank you for submitting your paper to PLOS one. I now have in hand two reviewer reports. While both reviewers are sympathetic to the aims and claims of your paper, they also find serious problems with the experimental design and data analysis. I find your article extremely engaging and I agree with you on the importance of the phenomena that it uncovers. I invite you to submit a revision of your paper that addresses these remarks, to which I add a few of my own in the below.

Like Reviewer 2, I found it impossible to replicate the reported results based on the data that was provided — speaking not just of the statistics but of the basic description of the results. The most obvious discrepancy is the number of participants: the paper (and S1) reports 64 participants, but the data lists only 62. In this connection I note that you did not preregister an exact number of participants, just noting that there would be between 15 and 20 per condition, with no rule provided on when to stop data collection. Please explain the discrepancy between the data and reported results, and provide an account of the rationale for stopping data collection at 16 (?) participants per condition. Of course all the other points raised by Reviewer 2 in connection with the data should be addressed too.

On a more theoretical note, the discussion of vertical vs. horizontal transmission is difficult to connect with your experiment, in which vertical vs. horizonal transmission is not at all manipulated. Reading this also reminded me of the numerous critiques that were levelled at Cavalli-Sforza et al.'s typology of transmission modes in the early 1980s, starting with Boyd & Richerson and continuing today. These critics took issue, among other things, with the claim that horizontal transmission can only sustain transmission for one generation. This is true only under very unrealistic modelling assumption, namely, if transmission is purely and perfectly horizontal, with no age difference whatsoever between the source and target. In reality, of course, this is never the case. In real life "horizontal transmission", as studied by anthropologists, there is always some age difference between the source and the target. The source might be 8 and the target 6. Technically speaking, all transmission is somewhat 'oblique' (if we really need to use this terminology). An important consequence of this is that culturel transmission between children can and does sustain long-standing cultural traditions, as research on children's folklore has solidly established (see also my own work on this). The claim that horizontal transmission cannot sustain cultural transmission over more than one generation is thus false under any realistic empirical interpretation of horizontal transmission.

Please consider these concerns and, even more importantly, the concerns raised by the reviewers, when revising your study. Please bear in mind this standard caveat if and when you revise the paper: Inviting a revision does not entail that the next version, or any subsequent version, will be accepted for publication. It is my policy to avoid a protracted editorial process that may in any case end in rejection. I am not pre-judging this particular case but this is something I warn all authors of.

We look forward to receiving your revised manuscript.

Kind regards,

Olivier Morin

Academic Editor

PLOS ONE

Journal Requirements:

Reviewers' comments:

Reviewer's Responses to Questions

**Comments to the Author**

1. Is the manuscript technically sound, and do the data support the conclusions?

Reviewer #1: Partly

Reviewer #2: Partly

2. Has the statistical analysis been performed appropriately and rigorously? 

Reviewer #1: Yes

Reviewer #2: No

3. Have the authors made all data underlying the findings in their manuscript fully available?

Reviewer #1: Yes

Reviewer #2: Yes

4. Is the manuscript presented in an intelligible fashion and written in standard English?

Reviewer #1: Yes

Reviewer #2: Yes

5. Review Comments to the Author

Reviewer #1: General Comments

1. The central claim about associative learning requires further clarification. As far as I can tell, the participants only undergo a single trail (acquisition -> transmission). This kind of design is insufficient to generate new associative effects. At best it can rely on preexisting cues/triggers about one’s social position within well-recognized social roles.

2.More generally, the design is not sensitive enough to determine underlying mechanisms. There are a number of candidate alternatives that are not ruled out. Nothing in the test, for instance, rules out the recognition of overt cues of status that are used to explicitly infer which strategy to deploy.

3. Along similar lines, I do not find the specific claim about context-congruence supported. To determine associative effects between context, one requires numerous trials to test and control for various contextual factors that might be driving the pattern of behaviour. That kind of careful work is not possible with the current design. As such—even if one were to have good reason to believe associative effects were driving the pattern of results—there is little support in the current study that congruence between social roles is the driving factor (see, e.g. Q6 below).

4. The attribution of "expertise" is thin on the ground. Just designating someone an expert is insufficient for either the acquisition of expertise or the felt experience of being an expert (this is acknowledged on lns. 530-533). The latter seems particularly important, since one must assume for the claim of associative learning to go through that low-level feelings of expertise (rather than explicitly entertained ideas about social roles) are what drive the choice of onward transmission. The lack of any measure on this puts the results of this study into question.

5. Similarly for the recognition of “peers” and “novices”. There is no manipulation that distinguishes “novices” from “peers”. “Peers” were simply marked out as “participants just like you.” But nothing distinguished a novice as a novice nor peers as peers. The artificiality of the experimental set up makes one further wonder the extent to which effects of "peer" identification are at work. That the authors themselves note these roles/identifications are "implied" (ln. 537) is unsatisfactory.

6.Further problematizing the design of the experiment was the choice of task. The task is both easy and causally transparent. This means that any number of effects (familiarity with the task, perceived ease of communication/implementation) might explain the pattern of results. A more compelling experimental test would have made the test causally opaque to control for these confounds.

7. In the literature review, there are several places where work is cited about the tendency for certain information to be transmitted by a certain modality (e.g. “subsistence and childcare skills” through vertical inheritance). The paper needs to make clear when the citations and claims are made about forager (also called “hunter-gatherer”) groups, and when these citations and claims are made about contemporary non-forager populations. As things are now, the literature review is misleading.

8. I don’t understand the lack of literature review on “transmitting onwards” (e.g. ln. 174-184). Where is this? There is a wealth of work in both CE and sociology on these claims. Consider that seminal texts of sociology from Pierre Bourdieu and Michèle Lamont concern how and why individuals decide to transmit and/or express the behaviors that they do. These quantitative and qualitative studies are a dime a dozen in sociology. Cultural evolution has jumped on this bandwagon. Recently, Hugo Mercier has summarized a wealth of theoretical and experimental work on this material in his Not Born Yesterday. This gap in the literature review needs to be addressed and carried through to the discussions and conclusions (e.g. lns 446-467).

Specific Comments

ln. 46: "through" not "though"

lns. 60-61: This isn't strictly true. Richerson and Boyd characterize content-based biases as capturing the selective retention of traits otherwise copied.

ln. 77: The language here is non-standard. Usually these are called "transmission modes", moreover, I do not think these modes attribute "different characteristics" to learners—only different dynamics to the overall population.

ln. 86: I don't know how to understand the claim that vertical inheritance is the "most adaptive transmission pathway." The claim seems ill-formed. Even if vertical inheritance dominates during adolescence, this doesn't mean it is the "most adaptive."

lns 98 - 100. What is the difference between "cultural information" and "cultural traditions"?

lns 101 - 112. The literature might confuse "vertical" and "oblique", but that's no excuse to muddy the waters here. The examples given here are instances of "oblique" transmission, rather than "vertical".

ln. 217. Following on from above. This is an “oblique” pathway, not a “vertical” one.

Reviewer #2: In this paper, Papa et al report the results from an experiment that teases apart the acquisition of cultural variants and the onward transmission of these variants. The general hypothesis is that there is a higher probability of producing variants when the contexts of transmission in the acquisition and onward transmission phases are congruent. The contexts here refer to learning situations that are either expert-to-novice or peer-to-peer. They find that participants are generally more likely to transmit the variant produced by the expert. However, this effect is modulated by the context-congruence: participants overwhelmingly transmitted the expert strategy to a novice, but less so to a peer.

One issue for me was the lack of clarity over the various theoretical concepts, the hypotheses generated, and the experimental predictions. Whilst the authors provide a thorough discussion of transmission pathways (e.g., vertical versus horizontal), model-based biases (which, in this case, is basically the level of perceived expertise), and associative learning, it is left to the reader to connect the dots when linking these to the hypotheses on lines 202 to 229. You state that your “first hypothesis concerned the model-based bias”, but then you don’t explicate or advance an actual hypothesis. Instead, you jump straight to the predictions (i.e., that participants are more likely to transmit the expert’s strategy). The second hypothesis is clearer and advances the following predictions: that (i) variants from a peer (in the peer-to-peer context) are more likely to be transmitted to another peer (in a new peer-to-peer context) and (ii) variants from an expert (in an expert-to-novice context) are more likely to be transmitted to another novice (in a new expert-to-novice context). But it is less clear to me how these predictions follow from the excellent theoretical set up you developed. I think the paper will greatly benefit from explicitly motivating the hypotheses based on the literature.

However, even if we take your predictions for granted, this leads to a second issue in how you interpret your results. Specifically, your results mainly focus on the how expertise is modulated by context of onward transmission. But I don’t understand why you wouldn’t expect a similar result for the transmission to peers? This is what you predict based on your second hypothesis. So, here, you would expect that a participant who is a peer in the acquisition phase would have a disproportionate tendency to transmit to a peer in the onward transmission phase? This sort of looks like the case in figure 4 if we only consider the right bar graph (onward transmission of peer’s strategy), but then we see that the participant-to-peer variant is at a similar level in the left bar graph (onward transmission to expert’s strategy). Another explanation for this is that the entire effect is just being driven by expertise: participant-to-novice is amplified in the onward transmission of expert’s strategy and attenuated in onward transmission of peer’s strategy. To me, this interpretation does not seem consistent with your context-congruence and associative learning hypothesis (as I’ve understood it). It might be that you want to say there is a relationship between your first hypothesis (a model-based bias for expertise) and your second hypothesis (of context-congruence, but only when interacting with expertise), but again this is not clear from what you wrote in the paper.

My final concern is with respect to the statistical models. The first aspect to mention here is that you only shared your data, not your R code. This makes it difficult for me to fully assess your procedure. I would like to see the code made available in the next revision of the paper. (Apologies if this an oversight of mine and I missed it in the main text.) The second, and more important, aspect is that I did look at your data and I managed to reproduce the models. However, this raises two issues with what you reported in the paper. In section 3.2 GLMER, you report your model as including a random intercept for participant. I do not see how this can be the case when each row in your dataset corresponds to a single participant (the ID variable in your dataset). I.e., for a random intercept to be informative here you would need more than one row per participant (which, in most experiments, would correspond to a participant doing multiple trials where each row is a trial). Moreover, your model produces singular fits even if I drop participant from the random effects, and indicates that your model is overfitted. In this specific instance, the overfitting is due to the complexity of your model relative to the sparsity of the data (you only have 62 datapoints in your dataset).

6. PLOS authors have the option to publish the peer review history of their article (what does this mean?). If published, this will include your full peer review and any attached files.

Reviewer #1: No

Reviewer #2: No

---

## [Author Response · Author response to Decision Letter 0]

15 Dec 2022

We have included a point-by-point response to all of the reviewers' comments (see attached letter).

---

## [Decision Letter · Decision Letter 1]

7 Feb 2023

PONE-D-22-12845R1Context congruence: How associative learning modulates cultural evolutionPLOS ONE

Dear Dr. Papa,

Thank you for submitting your manuscript to PLOS ONE. After careful consideration, we feel that it has merit but does not fully meet PLOS ONE’s publication criteria as it currently stands. Therefore, we invite you to submit a revised version of the manuscript that addresses the points raised during the review process. **Please see below for my editorial comments (OM)**

We look forward to receiving your revised manuscript.

Kind regards,

Olivier Morin

Academic Editor

PLOS ONE

Journal Requirements:

Additional Editor Comments:

Dear Monica,

Thank you for submitting a revision of your paper to PLOS. One of the two original reviewers accepted to review the revision and found that it addressed most of their recommendations satisfactorily. They note some lingering concerns, however. Please consider these points when revising the paper, the first two in particular (about GLMER and Bonferroni corrections).

I took note of your answer to my own recommendations too, but in this case I do not think the revised version addresses all of them. I am therefore asking you to revise your submission in line with two simple recommendations. You wrote in your response that the decision to run 64 participants "was taken in advance of data collection, and is in the pre-registered report. We planned to run 64 participants and only looked at the results after data collection had been completed." I reread the preregistration document, which just sets a minimum figure but no maximum. Please explain this in the paper. (I know I sound picky, but one of the important goals of preregistration is to prevent people from adding participants in arbitrary ways. Committing to a minimum number of participants only does not achieve that.)

On a more important point, I still see a big gap between the peer-to-peer vs. expert-to-novice comparison that is implemented in your experiment, and the distinction between horizontal and vertical/oblique transmission that you push in the introduction and conclusion. The two phenomena are distinct even though they may overlap. I very often get taught by younger experts on topics on which I am a novice. There may be a correlation between age and expertise, in some areas, but this does not mean they can be treated as equivalent. Please rewrite the paper to acknowledge clearly that expert-to-novice transmission is not the same thing as vertical/oblique transmission, and that, as a result, your expert-to-novice condition is not a vertical transmission condition.

Please accompany your resubmission with a track-changes version of the ms. (The submission you sent us did not have that. It contained a commented version of the previous ms with a general description of the changes, but PLOS requires a track-changes version showing all the changes that were made. Use "compare documents" if you are on Word.)

If satisfied with the changes, I will not be sending the revision back for another round of reviews.

Thank you again for sending us this engaging and thought-provoking study. I am looking forward to the revision.

Kind regards,

olivier

Reviewers' comments:

Reviewer's Responses to Questions

**Comments to the Author**

1. If the authors have adequately addressed your comments raised in a previous round of review and you feel that this manuscript is now acceptable for publication, you may indicate that here to bypass the “Comments to the Author” section, enter your conflict of interest statement in the “Confidential to Editor” section, and submit your "Accept" recommendation.

Reviewer #2: All comments have been addressed

2. Is the manuscript technically sound, and do the data support the conclusions?

Reviewer #2: Yes

3. Has the statistical analysis been performed appropriately and rigorously? 

Reviewer #2: No

4. Have the authors made all data underlying the findings in their manuscript fully available?

Reviewer #2: Yes

5. Is the manuscript presented in an intelligible fashion and written in standard English?

Reviewer #2: Yes

6. Review Comments to the Author

Reviewer #2: I would like to thank the authors for taking into consideration the comments; the paper is much clearer following the rewrite. Most of the issues I raised have now been addressed. However, I do have some lingering concerns, mainly to do with the statistical analyses.

As you note in your cover letter, you no longer use a GLMER in your paper and instead take two approaches: (i) a series of Chi-Square tests and (ii) a parameter estimation simulation. First, I think you will need to note why you decided to not use the GLMER model you pre-registered. I understand that this looks cumbersome in a paper, but perhaps you could update your OSF pre-registration to note this change.

Second, I noticed you did multiple chi-squared tests but did not correct for multiple comparisons. I noted 8 comparisons in your R analysis, which, if we use a basic Bonferroni correction, means that a P value must be less than 0.05/8=0.00625 to be significant at the standard P<0.05 level. Maybe you have a good reason for not doing such corrections, but currently it is not clear to me why this is the case.

Finally, I appreciate the use of simulations in the exploratory analyses, but I have two minor points to make here. A downside of your approach is that we cannot really capture the uncertainty of the parameter estimate. There are other methods for robustly inferring parameters, which are nevertheless closely related to your approach here, where you can try and estimate this uncertainty. One such method is to use Approximate Bayesian Computation (for an example applied to experimental data, see Vasishth, 2020). I am not asking you to implement this method. It would be useful though to acknowledge the limitations of the simulation approach you’ve adopted here. My second minor comment is that you report one sample t-tests, but you might want to motivate the use a bit more (and also to not report p-values = 0.000). It is not to say this approach is wrong; rather, as with many of these approaches, it is subject to considerable debate in the literature (see White et al., 2013).

Vasishth, S. (2020). Using approximate Bayesian computation for estimating parameters in the cue-based retrieval model of sentence processing. MethodsX, 7. https://doi.org/10.1016/j.mex.2020.100850.

White, J.W. et al. (2013). Ecologists should not use statistical significance tests to interpret simulation model results. OIKOS, 123:4. https://doi.org/10.1111/j.1600-0706.2013.01073.x

7. PLOS authors have the option to publish the peer review history of their article (what does this mean?). If published, this will include your full peer review and any attached files.

Reviewer #2: No

---

## [Author Response · Author response to Decision Letter 1]

18 Feb 2023

Dear Olivier,

Thank you for your and the reviewer’ comments on our resubmission. We believe we have addressed all the issues raised. Responses to each point are in the table below. 

With best wishes, 

Monica, Aliki, Mioara and Nicola

---

## [Editor Report · Decision Letter 2]

23 Feb 2023

Context congruence: How associative learning modulates cultural evolution

PONE-D-22-12845R2

Dear Dr. Papa,

We’re pleased to inform you that your manuscript has been judged scientifically suitable for publication and will be formally accepted for publication once it meets all outstanding technical requirements.

Kind regards,

Olivier Morin

Academic Editor

PLOS ONE
---

## [Editor Report · Acceptance letter]

23 Mar 2023

PONE-D-22-12845R2 

Context congruence: How associative learning modulates cultural evolution 

Dear Dr. Papa:

I'm pleased to inform you that your manuscript has been deemed suitable for publication in PLOS ONE. Congratulations! Your manuscript is now with our production department. 

Kind regards, 

on behalf of

Dr. Olivier Morin 

Academic Editor

PLOS ONE